# A Low Cost Inkjet-Printed Mass Sensor Using a Frequency Readout Strategy

**DOI:** 10.3390/s21144878

**Published:** 2021-07-17

**Authors:** Bruno Andò, Salvatore Baglio, Vincenzo Marletta, Ruben Crispino

**Affiliations:** Department of Electric Electronic and Computer Science Engineering (DIEEI), University of Catania, 95125 Catania, Italy; salvatore.baglio@unict.it (S.B.); vincenzo.marletta@dieei.unict.it (V.M.); ruben.crispino@unict.it (R.C.)

**Keywords:** mass sensor, InkJet Printing technology, modeling, characterization

## Abstract

The development of low-cost mass sensors is of unique interest for the scientific community due to the wide range of fields requiring these kind of devices. In this paper, a full inkjet-printed mass sensor is proposed. The device is based on a PolyEthylene Terephthalate (PET) cantilever beam (operating in its first natural frequency) where a strain-sensor and a planar coil have been realized by a low-cost InkJet Printing technology to implement the sensing and actuation strategies, respectively. The frequency readout strategy of the sensor presents several advantages, such as the intrinsic robustness against instabilities of the strain sensor, the residual stress of the cantilever beam, the target mass material, and the distance between the permanent magnet and the actuation coil (which changes as a function of the target mass values). However, the frictionless actuation mode represents another shortcoming of the sensor. The paper describes the sensor design, realization, and characterization while investigating its expected behavior by exploiting dedicate models. The working span of the device is 0–0.36 g while its resolution is in the order of 0.001 g, thus addressing a wide range of potential applications requiring very accurate mass measurements within a narrow operating range.

## 1. Introduction

In the last few decades, there has been a growing interest in disposable and low-cost sensors for several applications such as biochemical sensing in medicine, chemistry, and physics [1,2,3,4,5,6]. Such interest is justified by the increasing need for devices that can efficiently work in hostile environments or can suffer irreversible (e.g., chemical or physical) processes that could permanently compromise their functionalities. These applications require low-cost sensing elements realized by cheap materials which are compatible with addressed applications, especially when contaminants, irreversible processes, or harsh environments are involved.

In particular, the development of low-cost mass sensors is of unique interest for the scientific community, due to the wide range of fields requiring this kind of devices.

Within the framework of mass sensors, Bulk Acoustic Wave (BAW) sensors appear to represent a reliable solution, especially in addressing the development of devices featuring high sensitivity and resolution. These are based on piezoelectric resonators with suitable coatings exploiting the shift of resonant frequency due to the target mass effect [1]. Among BAW sensors, Quartz-Crystal Microbalances (QCM) are largely diffused [2,3]. The main limitations of QCMs are related to the need for thin crystals to increase the sensitivity and lack of selectivity, which require the integration of an array of sensors. Surface Acoustic Wave (SAW) sensors can represent a good solution to these kinds of sensitivity needs [4,5].

It is well known that many mass sensors are realized by measuring the strain of a membrane due to its target mass or by exploiting its effect on the resonance frequency of an inertial structure. In terms of readout strategies, piezo-resistive, capacitive, and inductive approaches are commonly proposed. The piezo-resistive strategy is convenient in terms of responsivity and also considering that, in case the resonant operation mode is adopted, drift and instability issues do not even affect the sensor performances.

A real-time mass sensor utilizing a Micro-Electro-Mechanical-System based self-sustained thermal-piezoresistive oscillator fabricated by a standard Silicon-On-Insulator process is presented in [6]. Examples of above-referred mass sensors for detection of albumin, glucose and yeast cells are given in [7,8]. The fabrication of a mechanically piezo-flexible load sensor exploiting multi-walled carbon nanotubes and simple carbon ink as the sensing components is presented in [9]. The realization of an optomechanical sensor for mass measurements is reported in [10]. This device uses a pair of optical fibers and a reflecting coated lens. The light intensity modulation is based on the relative motion of the lens due to a given mass with respect to the optical fibers. 

The fabrication, packaging and testing of a resonant mass sensor for the detection of biomolecules in a microfluidic format are presented in [11]. A highly sensitive, Film Bulk Acoustic Resonator (FBAR) mass sensor (built on a micromachined silicon-nitride diaphragm with a piezoelectric thin film and aluminum electrodes) that can operate in vapor and liquid is proposed in [12]. A high quality-factor silicon cantilever-based mass sensor has been investigated in [13], while a capacitive mass sensing with a 250-nm-thick single-crystalline silicon cantilever has been investigated in [14]. Lead Zirconate Titanate (PZT) films screen-printed on alumina substrate form composite piezoelectric resonators, which can be exploited as acoustic-wave mass sensors, are discussed in [15]. A self-oscillating MEMS mass sensor with a uniform mass sensitivity to directly measure the mass changes of evaporating microdroplets is presented in [16]. In [17], a resonant mass sensor is proposed which exploits ferrofluid to implement actuation and readout strategies. Other valuable examples of the use of QCM and SAW in frontier applications are given in [18,19,20,21].

The above papers demonstrate the feasibility of mass sensors with high performances at the expense of demanding realization technology as well as complex sensor topologies. 

The idea proposed throughout this paper is the realization of a low-cost sensor in the mesoscale, exploiting a disposable sensing element realized by using a flexible substrate and low-cost technology. Joining the need for low-cost devices, flexible substrates and mesoscale development, printed technology, and plastic/polymeric substrates could represent a convenient choice for the realization of mass sensors [22,23,24,25]. 

Recently, the scientific community has shown a growing interest in the possibility of developing cheap flexible electronics by exploiting innovative materials and printing technologies. This interest is driven by several reasons such as the need for low-cost mass-production processes, applications requiring shapeable and disposable devices, and the need of fast prototyping of electronics and sensors, which is of great interest for the scientific community as a whole.

Among printing technologies two main classes can be identified. The first one includes all techniques requiring mask-based or photolithographic processes (e.g., screen printing and printed circuit board (PCB) technology). The second class is related to direct printing technologies (e.g., InkJet Printing (IJP), Micro-Plotter, Aerosol Jet printing, and ElectroHydroDynamic (EHD) Jet Printing). A well elaborated review of digital printing technology is available in [26].

Direct printing technologies, particularly in regards to IJP, have unique advantages such as high spatial resolution, compatibility with many substrates, and a contactless deposition technique. Moreover, with respect to photolithographic processes, direct printing approaches are mask-less, which enables the rapid prototyping of devices and inks’ saving.

InkJet Printing technology affords the possibility of combining performances of flexible substrates and functional inks with applications requiring the rapid development of sensors and electronic components. Moreover, the high throughput capability of modern printing heads allows for the scaling up of IJP for mass production.

The main limitation of IJP technology is due to the low viscosity value of printable inks, which may lead to non-uniform widths and thicknesses of printed patterns. Dedicated techniques can be used to print high viscosity inks, such as contact type dispensers (Time-pressure, piston based and rotary screw) or non-contact jet-type dispensers, exploiting the motion of a needle inside an injection chamber. 

In order to deposit high viscosity inks with high resolution, micro-plotter technology can be adopted, which uses a dispensing mechanism based on the ultrasonic pumping action [26]. Another non-contact direct printing technology is Aerosol Jet printing, which allows for the deposition of materials with a wide range of viscosities also over non-flat surfaces [27]. The EHD jet printing, which uses electrostatic forces to release very small drops in the order of few micrometers, represents a suitable solution for those applications requiring patterns printed with a very high resolution [28].

When considering IJP equipment, several options are available on the market, including professional printers and low-cost printers. Nowadays, low-cost IJP solutions are under consideration, thus addressing the need of realizing lab-scale prototyping (especially for research and educational purposes [29,30,31]) requiring simple pre and post-processing.

In this paper, on the basis of the above needs and technological considerations, a full inkjet-printed mass sensor is proposed. The device is based on a PET cantilever beam, operating at its first natural frequency, where the strain-sensor and the actuation coil have been realized by a low-cost IJP technology.

As far as we know, the solution proposed in this paper is the first example of a low-cost, fully-printed mass sensor adopting a robust frequency readout strategy. The frequency readout strategy of the sensor presents several advantages, such as the intrinsic robustness against instabilities of the strain sensor, the residual stress of the cantilever beam, the target mass material, the distance between the permanent magnet, and the actuation coil. The latter changes as a function of the target mass values. Moreover, the contact-less actuation mode is another peculiar shortcoming of the proposed sensor.

Another important outcome of this work is related to the device modeling which, as discussed in the paper, allows for the design of the whole system, in particular with regards to the actuation part.

The use of a IJP technology for the realization of the system shows several advantages. Being a rapid prototyping techniques, it allows one to implement a convenient develop and test approach which is strategic for investigating the suitability of the proposed methodology by the fast development of Lab-Scale prototypes. Moreover, IJP allows for the easy realization of planar structures on flexible substrates (e.g., the strain gauge and the coil in the developed prototype). Finally, the low-cost feature of the proposed solution is also interesting. The latter arises from the technological approach adopted for the realization of the sensing and actuation strategies (the flexible PET cantilever beam and the low-cost inkjet-printed devices).

The working span of the device is 0–0.36 g, while its resolution is in the order of 1.00 mg, thus addressing applications requiring very accurate mass measurements within a narrow operating range. This represents a further interesting advantage of the proposed solution.

In the following sections, the working principle of the sensor is presented, along with considerations leading to sensor modeling and design. The full characterization of the sensor is discussed in Section 3. Finally, conclusions are drawn in Section 4.

## 2. The Device Developed

### 2.1. Working Principle and the Device Components

The schematization of the mass sensor developed is illustrated in Figure 1. The device consists of a PET cantilever beam, which is driven to its first natural mode by an electromagnetic forcing system. In particular, the forcing mechanism exploits the interaction between a permanent magnet, placed at a distance d from the beam, and a coil that has been inkjet-printed on the end part of the beam. The magnet position can be modified to investigate the system behavior for different values of the force applied to the beam. Target masses are positioned close to the beam end. The beam is forced with an impulse signal which brings the structure to its oscillating regime. The sensing mechanism is based on the variation of the beam natural frequency (more specifically, its first natural frequency) as a function of the target mass. The beam dynamic is observed by a strain gauge, which has been inkjet-printed on the cantilever beam close to the fixed-end. 

To cope with specific applications requiring accurate measurements of small masses, the operating range of the device has been fixed in the range of tens of milligrams. The beam dimensions have been fixed by the application, as discussed in the next section.

In the following sections, details of the technology and materials adopted for the realization of the inkjet-printed devices are given, along with the other system components including the electronics adopted for the actuation system and the readout strategy.

#### 2.1.1. The Magnetic Field

The permanent magnet adopted is the Neodymium, N42, nickel-plated Q-15-15-08-N, which was chosen to be compliant with the structure dimensions.

The following model has been used to estimate the flux density of the permanent magnet close to the inkjet-printed coil:(1)$$B=\frac{B_{r}}{\pi }\left[atan\left(\frac{L_{M}W_{M}}{2d\sqrt{{4d}^{2}{+L}_{M}^{2}{+W}_{M}^{2}}}\right)-atan\left(\frac{L_{M}W_{M}}{2(D+d)\sqrt{{4(D+d)}^{2}{+L}_{M}^{2}{+W}_{M}^{2}}}\right)\right]$$
where:

*B_r_* is the remanence field in Gauss;

*L_M_* is the length of the block;

*W_M_* is the width of the block;

*D* is the thickness of the block; and

*d* is the distance from a pole face on the symmetry axis.

To estimate the remanence field, *B_r_*, an experimental survey has been performed by measuring the magnetic field while changing the distance, *d*, between the magnet and the sensor, placed one in front of the other. For such an aim, the Hall effect sensor HAL2455 by Micronas has been used. The latter is a high-precision linear Hall-effect sensor with 12-bit resolution and PWM output up to 2 kHz that permits magnetic field measurements in the range up to 200 mT (2000 G).

Figure 2 shows the measured values of *B* as a function of the distance d and the expected trend obtained by fitting model (1) onto the experimental data. A value of *B_r_* = 12,750 G has been estimated, which falls within the expected range provided by the magnet manufacturer. The fitting process has been performed by using the Nelder–Mead optimization algorithm [32], implemented through a dedicated Matlab script.

The two extremes of the magnetic field have been set to guarantee a suitable operating range of the device. With the aim of investigating the system behavior as a function of the applied magnetic field, two values of the d distance will be considered in the following, 7 mm and 10 mm, which produce magnetic field strength of 1617 G (0.1617 T) and 948 G (0.0948 T), respectively.

#### 2.1.2. The Cantilever Beam and the Inkjet-Printed Strain Gauge

Figure 1b shows the layout of the inkjet-printed elements, including the actuation coil and the Strain Gauge (SG) implementing the readout strategy.

Devices have been realized on a PET substrate, with a thickness of 120 µm, by using a low-cost EPSON piezo inkjet printer. The silver nano-particle solution “Metalon^®^ JS-B15P” by Novacentrix has been chosen, being compatible with the selected printing machine. Curing post-processing has been performed at 90 °C for 1 h.

The adopted printing technology does not allow for controlling the ink layer thickness. Electron microscopy (SEM) analysis of the silver pattern deposited on the substrate has shown a homogeneous layer with a thickness of about 1.90 µm.

Dimensions of the beam are given in Figure 1b. The beam’s width, *W_b_*, is fixed to cope with the needs of properly fixing the target mass on the sensing area and to design a suitable strain sensor. The cantilever length, *L_b_*, has been fixed to allow the realization of both the strain sensor and the actuation coil, according to constraint provided by the adopted IJP technology. 

The expected first natural frequency of the beam is around 10 Hz, as estimated through the following model [33]:(2)$$\omega_{0}=\frac{\alpha_{1}^{2}}{2}\sqrt{\frac{{ET}_{b}^{2}}{{3\rho L}_{b}^{4}}}$$
where:

$\alpha_{1}=1.875$ is a known coefficient known from the literature [33];

$E=2.0\times 10^{9}\;N/m^{2}$ is the Elastic module of the PET beam;

*T_b_* = 120.0 µm, is the PET thickness; and

*ρ* = 1.38 × 10^3^ kg/m^3^ is the PET density.

The strain sensor has been designed to capture the main curvature area of the beam. In order to optimize the sensor responsivity, the track spacing and width adopted for the SG realization are the minima allowed by the adopted technology, 300 µm and 200 µm, respectively.

The gage factor of the printed strain gauge, *GF_ijp_*, has been estimated by an independent measurement of the strain response to an impulsive stimulation. This measurement has been performed by a commercial SG bonded on the opposite side with respect to the printed SG. The following model has been used to fit the strain response, *ε_obs_*, observed by using the commercial SG, in the case of the impulsive input:(3)$$\varepsilon_{obs}=\frac{4\;V_{out}}{{GF}_{ijp}V_{b}}$$
where: 

*V_out_* is the output signal provided by the readout electronics used for the inkjet-printed SG, compensated by the amplifier gain; 

*V_b_* = 1.0 V is the supply voltage of the adopted Wheatstone bridge. More details on the conditioning electronics are given in the next section.

The fitting procedure, performed through the Nelder-Mead optimization algorithm [32], leads to an estimation of *GF_ijp_* equal to 1.68. The measured resistance value of the inkjet-printed SG is *Ro* = 226 Ω.

#### 2.1.3. The Inkjet-Printed Coil and Modeling of the Actuation System

The coil design, in terms of the number of turns and the coil track width, *W_c_*, is strictly constrained by:

-the beam dimensions, with particular regards to the actuation section of the beam (close to the beam free end); -the required current to produce, in the whole range of the sensor operation (tens of milligrams), a suitable beam deflection.

To such aim, the behavior of the actuation system has to be investigated, taking into account the minimum measurable strain, compared to the nominal resolution of the readout electronics, as well as the maximum beam deflection compatible with the design constraint.

The minimum measurable strain has been estimated by considering the noise floor observed, *δV*, in the output voltage of the resistive readout strategy:(4)$$\delta \varepsilon =\frac{4\;\delta V}{V_{b}{GF}_{ijp}}=67\;\mu \varepsilon$$
where: 

δV = 28.2 µV has been experimentally estimated, by observing the output response of the conditioning electronics in the absence of stimulation.

The maximum allowed deflection is 7 mm in case of *B* = 0.1617 T and 10 mm in case of *B* = 0.0948 T.

With reference to the coil layout in Figure 1b, the following quantities are fixed by the adopted technology:-External Guard Ring, *D_G_* = 1.5 mm;-Internal Coil Diameter, *D_in_* = 2.0 mm;-Coil track Spacing, *S* = 500 µm

The number of turns, *N*, can be defined as a function of above defined quantities:(5)$$N=int\left(\;\frac{W_{b}-{2D}_{G}\;-\;D_{in}-{2W}_{c}}{{2W}_{c}+2S}+1\right)\;$$

To estimate the optimal coil geometry, in the following a model describing the sensor behavior as a function of the driving current, the magnetic field, and the target mass, is introduced.

In particular, the target is to fix the optimal combination among the coil width, the driving current and the magnetic field, which can assure the minimum measurable strain in case of a null mass and the maximum beam deflection in case of the maximum expected mass value.

Since the operating mode of the device exploits the estimation of the natural frequency by the impulse response, the above estimations will be developed by considering the maximum value of the impulse response, which is related to the maximum value of the applied impulsive force.

The magnetic force, *F_m_*, acting on the beam structure is given by:(6)$$F_{m}{=k*I}_{d}{*L}_{tot}*B$$
where:

*I_d_* is the driving current flowing through the coil;

*B* is the magnetic field;

*k* is a fitting parameter taking into account the non-ideal coupling between *B* and *I_d_*. A *k* value of 0.1 has been obtained in previous works by fitting the model (6) to experimental data [34]. Such a value of *k* has been then confirmed by fitting the model (6) on the observed beam strain, as discussed in Section 3.

*L_tot_* is the total coil length.

The overall force acting on the beam is given by:*F* = *F_m_* + *M**g*(7)
where *M* is the target mass and *g* the gravity acceleration.

In order to estimate the deflection at the free cantilever beam, *D*, and the strain, *S*, the following relationships have been used [33]:(8)$$D=\frac{{F*L}_{b}^{3}}{3*E*J}$$
(9)$$S=\frac{{6*F*L}_{b}}{{E*W}_{b}{*T}_{b}^{2}}{+S}_{0}$$
where:

*S*_0_ = 500 με is the offset strain measured at the free beam end in case of null *I*, *B* and M;

$J=\frac{W_{b}T_{b}^{3}}{3}$ is the moment of inertia.

Results obtained by simulating models (6)–(9) are presented in Figure 3. In particular, the driving force, the strain and the displacement of the beam have been estimated, as a function of the driving current, *I_d_*, for different coil track width and two values of the target mass, 0 and 3.6 g, in case of the two considered *B* intensities.

As it can be observed, the coil track width of 2 mm is the minimum assuring a good compromise between the desired driving force intensity (producing a substantial readable strain of the beam and a deflection compliant with the structure geometry) and the possibility to investigate the sensor behavior for driving current values up to 100 mA.

To better investigate the expected system behavior, Figure 4 shows the beam strain and deflection in case of a track width of 2 mm, as a function of the target mass, for different values of *I_d_* and the two values of *B*.

Concluding, as it can be observed by results provided in Figure 3 and Figure 4, constraints on the minimum beam strain and maximum deflection are fulfilled in the whole investigated range of *M*, *I* and *B*.

### 2.2. The Real Sensor and the Conditioning Electronics

On the basis of outcomes provided by simulating models (6)–(9), the coil track width has been fixed to 2 mm. A real view of the sensor is given in Figure 5.

The electromagnetic actuation system of the mass sensor is driven by a Voltage-to-Current Converter (VCC). In particular, the *OPA 547* Operational Amplifier has been used, due to its low-cost and high-voltage/high-current operation. The non-inverting configuration uses a power reference resistor of 10 Ω to generate the desired excitation current $i_{ex}\left(t\right)=\frac{Vex\left(t\right)}{10}$, where *Vex(t)* is the excitation voltage of the driving circuit. Such current flows through the printed coil, which is connected in the feedback chain with a reference shunt resistor used to perform an independent measurement of the excitation current. The conditioning electronics embedded in the readout sensing strategy is a Wheatstone bridge, supplied by a DC voltage, *V_b_* = 1.0 V. The resistance values have been fixed to maximize the circuit responsivity and to guarantee a maximum current flowing through the IJP strain gauge of 5 mA, which fulfills the sensor specification.

It must be bear in mind that, being the main information on target quantities contained in the oscillation frequency of the beam, the design of the conditioning electronics must be more focused on its performance in the frequency domain rather than in the amplitude domain.

The main target of this circuit is to convert the dynamic provided by the strain gauge into a readable voltage signal, which conveys the information on the natural frequency of the beam oscillation. To such aim the gain of the instrumentation amplifier has been experimentally fixed, by assuring a suitable behavior of the readout chain for the whole operating range of the sensor, taking into account the investigated range of driving current and excitation magnetic field. This approach represents one of the main advantages of the proposed approach, which is robust against the instabilities of the strain sensor, the residual stress of the cantilever beam, and the distance between the permanent magnet and the actuation coil, which indeed changes as a function of the target mass values.

Signals driving the actuation system and provided by the readout electronics are managed through a data acquisition board by National Instruments and a dedicated LabVIEW virtual instrument.

## 3. The Characterization of the System Prototype

In this section, results obtained by investigating the behavior of the mass sensor are presented as a function of target mass belonging to the range [0–0.36] g (9 calibrated masses of 0.04 g have been used), for different values of the driving current, *I_d_*, in the range [20–100 mA] and the two values of the magnetic field, *B*.

It must be observed that the small variation of the distance d, due to the target masses, produces a small variation of the actuation force applied to the beam. Anyway, this effect is negligible and will not affect the validity of the proposed methodology based on a frequency readout strategy (which is intrinsically insensitive against the driving signal strength).

The measurement protocol consists of applying ten current impulses, with a period of 4 s and a duty-cycle of 1%, while recording the strain at the fixed-beam-end measured through the IJP strain sensor.

Figure 6 shows the strain measured vs. the target mass, *M,* in the case of *B* = 0.1617 T and *I_d_* = 60 mA. As it can be observed, obtained values are compliant with the expected behavior, already investigated in Section 2. Moreover, fitting model (9) to observed data allows confirming a value of 0.10 for the fitting parameter, *k*. The minimization procedure has been implemented by using the Nelder–Mead optimization algorithm [32], while the Root Mean Square (RMS) error between the observed behavior and the model is 7.13 × 10^−6^.

Obtained signals have then been processed by computing their FFT to estimate the oscillation frequency. Figure 7 shows the behavior of the sensor for different values of the driving current and two values of the magnetic field. As it can be observed, the sensor response (in terms of natural frequency) is practically insensitive to the different operating conditions.

Based on the observed behavior, the most convenient actuation mode is the one exploiting the lowest magnetic field (allowing for a wider operating range) and current, being 0.0948 T and 20 mA, respectively.

The sensor response in the case of a magnetic field of 0.0948 T and a current of 20 mA is shown in Figure 8a. Resembling considerations leading to model (2), the following model has been used to interpolate the observed behavior of the beam natural frequency:(10)$$f_{beam}^{mod}=\alpha \sqrt{\frac{1}{{M+m}_{0}}}$$
where:

*α* is a constant taking into account the beam properties and estimated by fitting the model to the observed behavior;

*m*_0_ is the cantilever mass; and

*M* is the target mass.

For the sake of completeness, ten repeated readings per each value of the target mass are reported in Figure 8a (although not easily readable, due to their closeness), as well as the fitting model (10).

The fitting process, implemented through the Nelder–Mead optimization algorithm [32], allows estimating a value of 4.0 for the fitting parameter, *α*. The RMS error between the observed behavior and model (10) is 0.05 Hz.

The calibration diagram obtained in the case of a magnetic field of 0.0948 T and a driving current of 20 mA is shown in Figure 8b. The calibration function adopted has been estimated by inverting model (10). The sensor accuracy, given by the uncertainty bandwidth, *U_M_*, estimated in the 2σ limit, is equal to 8.0 mg.

In the following, considerations leading to the estimation of the sensor resolution are carried out. A first contribution to the sensor resolution is given by the resolution of the frequency estimation process which is:(11)$$\delta_{f1}=\frac{f_{s}}{N_{fft}}=\frac{5\;\mathrm{kHz}}{500,000\;}=\;0.01\;\mathrm{Hz}$$
where *f_s_* is the sampling frequency of the data acquisition system, while *N_fft_* is the number of samples used to estimate the FFT.

The other major contribution to be taken into account, *δ_f_*_2_, is due to the sensor output distribution, under the same operating conditions, in case of a null target mass. The estimated value of *δ_f_*_2_ is 0.008 Hz.

Above contributions can be combined as follows:(12)$$\delta_{f}=\sqrt{\delta_{f1}^{2}{+\delta }_{f2}^{2}}$$

Which reflects into the following sensor resolution:(13)$$\delta_{M}=\frac{\delta_{f}}{R}$$
where *R* is the device responsivity.

The responsivity and the resolution estimated by using the model (10), in the operating conditions, *B* = 0.0948 T and *I_d_* = 20 mA, are given in Figure 9. The Span-to-Resolution performance index of the sensor is shown in Figure 10.

As last, the Q factor of the IJP mass sensor has been estimated. To such aim the expected theoretical behavior of the beam has been fitted to behaviors observed for different values of the magnetic field and the driving current. This procedure allows for estimating a damping factor of the beam equal to 0.014 ± 8%, which leads to a Q factor for the device around 35. This result confirms the suitable performances of the sensor in terms of selectivity and resolution.

## 4. Conclusions

In this paper a full inkjet-printed mass sensor is proposed. The device exploits a PET cantilever beam, where a strain-sensor and a planar coil have been realized, by a low-cost InkJet Printing technology, to implement the sensing and actuation strategies, respectively.

The device exploits a frequency readout strategy, which offers several advantages, such as the intrinsic robustness against electrical noise.

The coil geometry has been optimized to allow desired performances in terms of the minimum readable beam strain and the maximum allowed beam deflection. The expected behavior of the sensor, in terms of the strain produced close to the fixed-end of the beam, has been investigated through dedicated models and successively confirmed by real observations.

A deep experimental analysis allowed to define the best operating conditions for the actuation strategy, both in terms of the driving current and the external magnetic field.

The trend for the resonant frequency of the sensor as a function of the target mass has been experimentally investigated, showing a strong coherence with theoretical predictions.

The estimated sensor span, resolution, and responsivity are compliant with real applications. In particular, the working range of the sensor is 0–0.36 g, while the resolution and the responsivity are in the range of 1.0 mg and 15 Hz/g, respectively. The Span-to-Resolution value, being around 400, and a Q factor of 35 confirm the suitable performances of the sensor in terms of selectivity and resolution.

Future efforts will be dedicated to assess the robustness of the mass sensor against small variation of the system parameters. As an example, the possibility of using professional printers, such as the Dimatix DMP2850 or similar systems, allowing to set the silver layer thickness will be taken into account. Moreover, the design of devices, exploiting the proposed sensing methodology, showing a wide operating range will be also addressed.

## Figures and Tables

**Figure 1 sensors-21-04878-f001:**
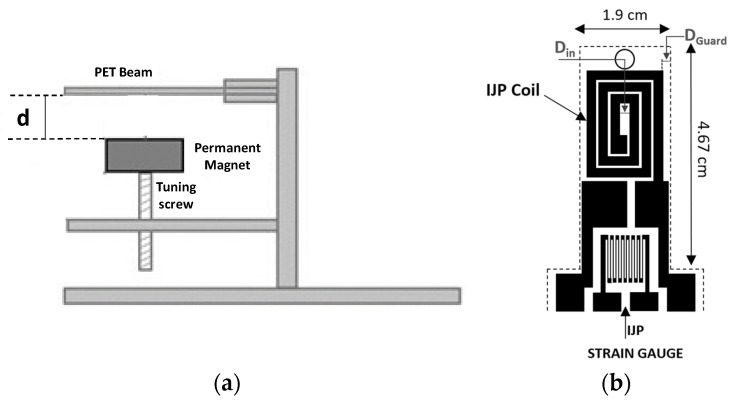
(**a**) Schematization of the mass sensor architecture and (**b**) layout of the inkjet-printed components.

**Figure 2 sensors-21-04878-f002:**
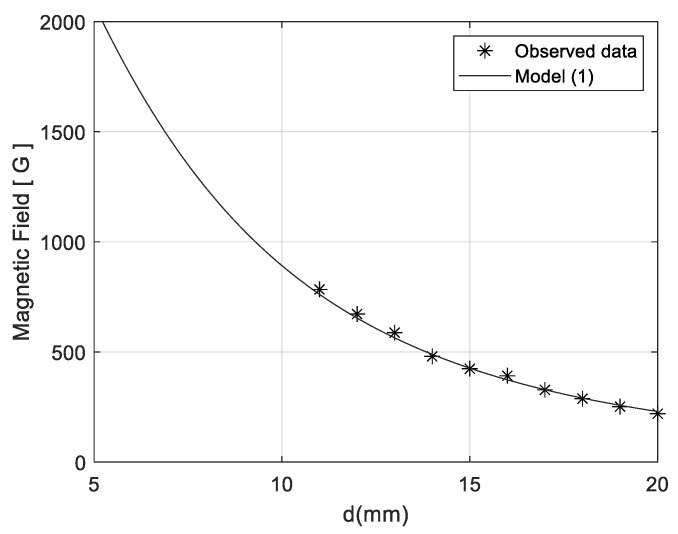
Measured and estimated magnetic field intensity as a function of the distance, d, from the permanent magnet.

**Figure 3 sensors-21-04878-f003:**
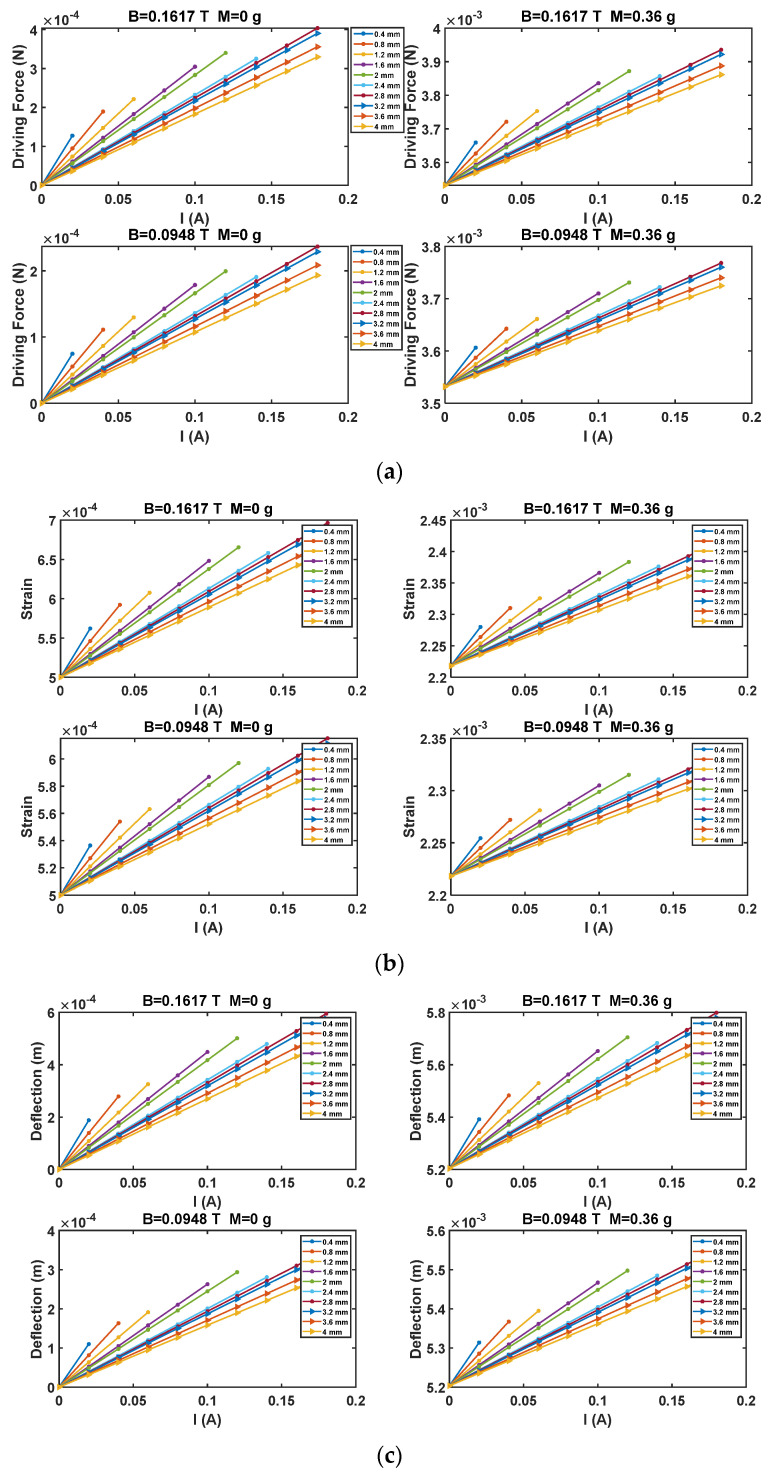
The actuation force (**a**), the strain (**b**), and the deflection (**c**) of the beam, as a function of the driving current, the coil track width, and two *B* intensities (0.0948 T, 0.1617 T). Results for two values of the target mass (0 and 0.36 g) are shown.

**Figure 4 sensors-21-04878-f004:**
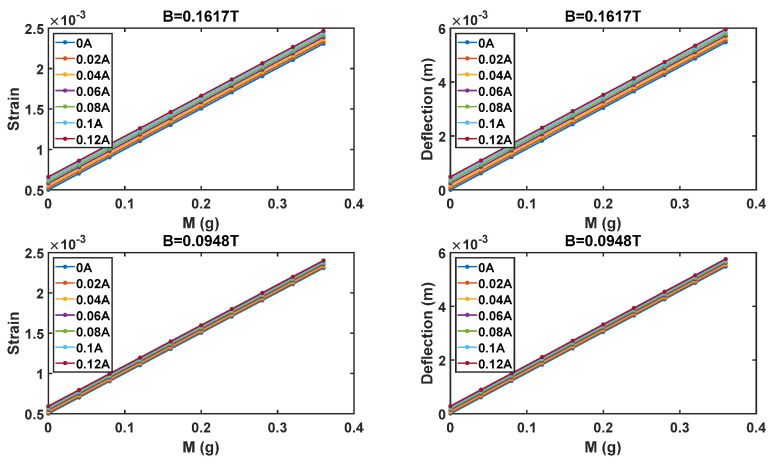
The strain and the deflection of the beam, as a function of the target mass and the driving current, in the case of two *B* intensities (0.0948 T, 0.1617 T).

**Figure 5 sensors-21-04878-f005:**
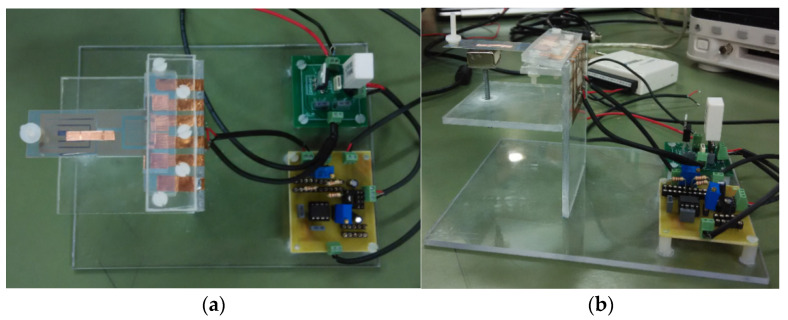
The developed mass sensor prototype. Top view (**a**) and lateral view (**b**).

**Figure 6 sensors-21-04878-f006:**
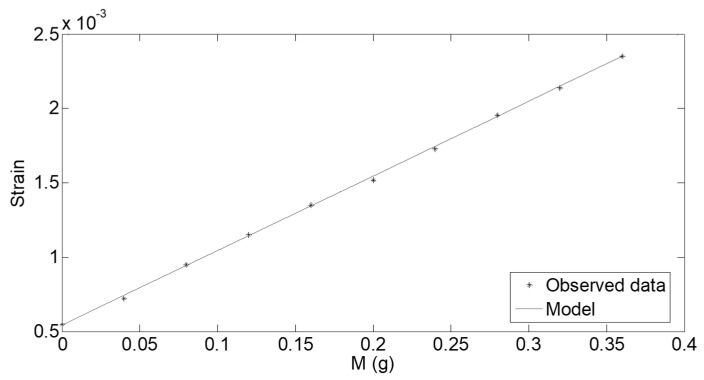
Fitting of the model (9) onto the observed strain in case of *B* = 0.1617 T and *I_d_* = 60 mA.

**Figure 7 sensors-21-04878-f007:**
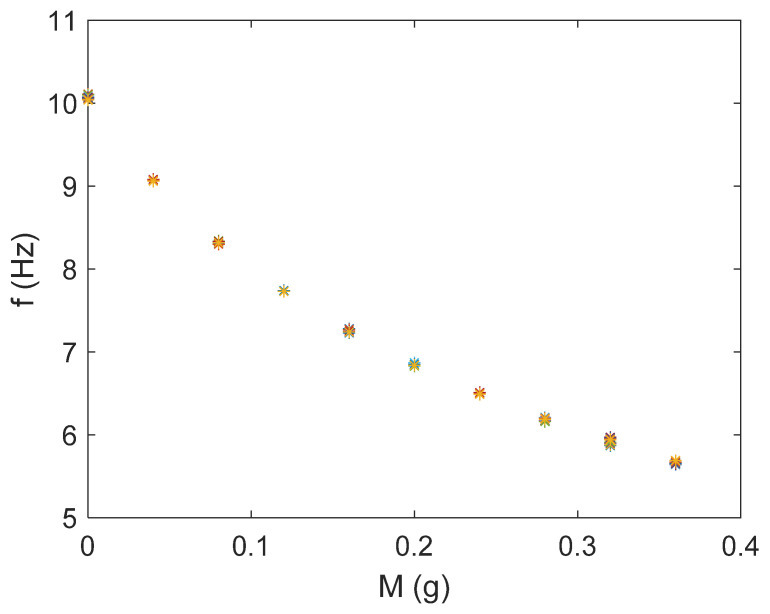
The sensor response for different values of the magnetic field, *B*, and the driving current, *I_d_*. Symbols represent different operating conditions. The device response is quite insensitive to *I* and *B*.

**Figure 8 sensors-21-04878-f008:**
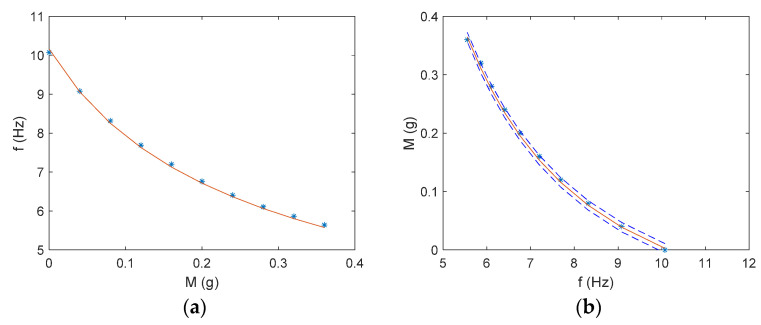
(**a**) Sensor response in the case of the following operating conditions: *B* = 0.0948 T and *I_d_* = 20 mA. Model (10) has been used to fit experimental data. (**b**) Calibration Diagrams for the same operating conditions, obtained by inverting model (10). A coverage factor of 2 has been used for the uncertainty bandwidth.

**Figure 9 sensors-21-04878-f009:**
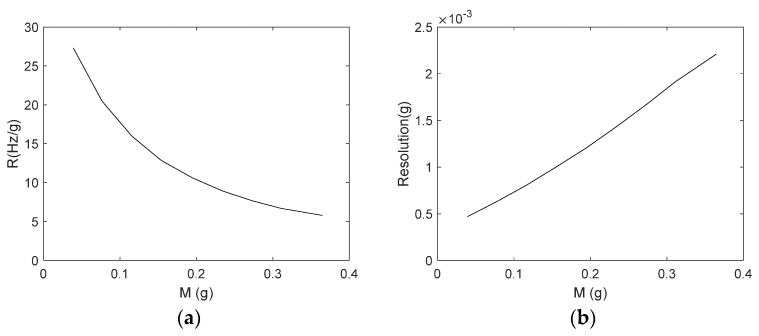
Responsivity (**a**) and resolution (**b**) of the mass sensor estimated in the following operating conditions: *B* = 0.0948 T and *I_d_* = 20 mA.

**Figure 10 sensors-21-04878-f010:**
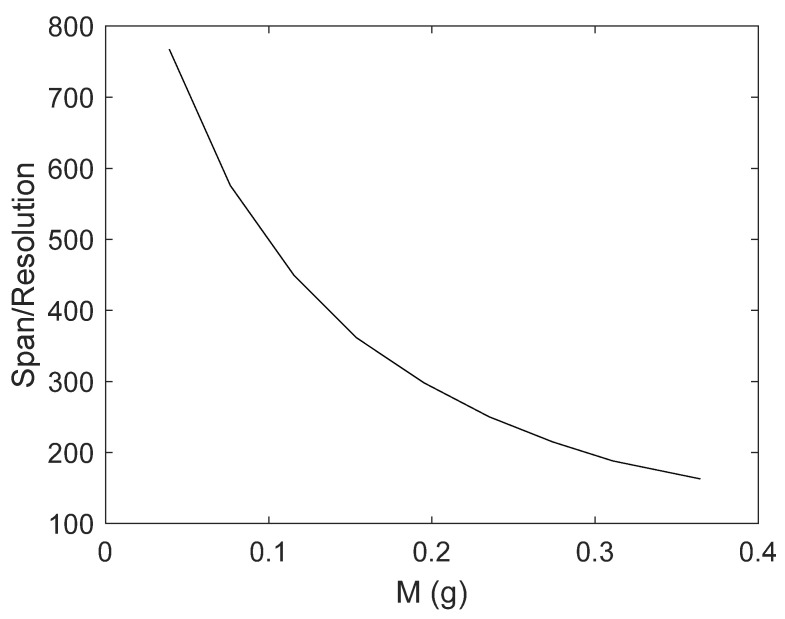
Span-to-Resolution of the mass sensor estimated in the following operating conditions: *B* = 0.0948 T and *I_d_* = 20 mA.

## Data Availability

Not applicable.

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
