# Peer review of "A Low Cost Inkjet-Printed Mass Sensor Using a Frequency Readout Strategy"

_sensors, 2021, doi:10.3390/s21144878_

Round 1

Reviewer 1 Report

State-of the art review on technology selection must be extended. You only mentioned a screen-printing and an inkjet-printing.

The section on inkjet printing technology  must be extended. It should data on thickness of inkjet-printed layer, width, and space.

Did you study impact of thickness? If yes, please include that. If not, please explain how you select that.

Same for the printed track width and space.

Did you try other silver ink? How did you select the ink?

How did you cure the inkjet printed features? What was the curing temperature and curing duration?

Author Response

REVIEWER 1

We would like to thank the Reviewer for the precious suggestions which allowed for improving the paper quality.

State-of the art review on technology selection must be extended. You only mentioned a screen-printing and an inkjet-printing.

The state-of-the-art has been extended by mentioning also other kind of direct printing technologies.

The section on inkjet printing technology  must be extended. It should data on thickness of inkjet-printed layer, width, and space.

Electron microscopy (SEM) analysis of the silver pattern deposited on the substrate has shown an homogeneous layer with a thickness of about 1.90 µm.

The track spacing and width adopted for the SG realization are the minima allowed by the adopted technology, 300 µm and 200 µm, respectively.

The coil track width and spacing are 2.0 mm and 500 µm, respectively.

Above information have been now included in the main text.

Did you study impact of thickness? If yes, please include that. If not, please explain how you select that.

The adopted technology, “low-cost EPSON piezo inkjet printer” does not allow for controlling the ink layer thickness. As above mentioned, this has been estimated by SEM analysis. Future efforts will be dedicated to investigate the effect of the ink thickness on the device performances, by using professional printers, such as the Dimatix DMP2850 or similar systems.

Above statements have been now reported in the main text.

Same for the printed track width and space.

Concerning the Strain Gauge, in order to increase the sensor responsivity, the choice was to choose the minimum track width and spacing allowed by the adopted technology.

Concerning the coil, the minimum track spacing have been fixed, according to constraints of the adopted technology, while the track width has been estimated by modeling the system behavior as discussed in Sect. 2.1.

Above statements have been now fully included in the main text.

Did you try other silver ink? How did you select the ink?

How did you cure the inkjet printed features? What was the curing temperature and curing duration?

The silver nano-particle solution “Metalon® JS-B15P”, by Novacentrix, has been chosen being the one compatible with the selected printing device. As declared in the paper, the curing post-processing has been performed at 90°C for 1 hour.

Motivation for using the selected ink has been now highlighted in the main text.

Reviewer 2 Report

The authors have reported a full inkjet-printed mass sensors based on cantilever beam.

    1. Authors should clearly compare the results of their work with the state of art. 
    2.  Authors should emphasize the advantage or impact of the methodology used in your work.
    3. Higher quality figures are required, especially Figures 7-10. 
    4. English and style are minor spell check required.

Author Response

REVIEWER 2

We would like to thank the Reviewer for the precious suggestions which allowed for improving the paper quality.

Authors should clearly compare the results of their work with the state of art.

Authors should emphasize the advantage or impact of the methodology used in your work.

As far as we know, the solution proposed in this paper is the first example of a low-cost, fully-printed, mass sensor, adopting a robust frequency readout strategy. The frequency readout strategy of the sensor presents several advantages, such as the intrinsic robustness against instabilities of the strain sensor, the residual stress of the can-tilever beam, the target mass material, the distance between the permanent magnet and the actuation coil. The latter changes as a function of the target mass values. Moreover, the contact-less actuation mode is another peculiar shortcoming of the proposed sensor.

Another important outcome of this work is related to the device modeling which, as discussed in the paper, allows for the design of the whole system, with particular regards to the actuation part.

The use of an inkjet printing technology for the realization of the system has several advantages. Being a rapid prototyping techniques it allows to implement a convenient develop&test approach, which is strategic for investigating the suitability of the proposed approach by the fast development of Lab-Scale prototypes. Moreover, IJP allows for the easy realization of planar structures on flexible substrates (e.g. the strain gauge and the coil in the developed prototype). As last, the low-cost feature of the proposed solution is also interesting. The latter arises from the technological approach adopted for the realization of the sensing and actuation strategies (the flexible PET cantilever beam and the low-cost inkjet-printed devices).

Above statements have been now better reformulated in the paper.

Higher quality figures are required, especially Figures 7-10. 

Figure style has been improved. Original figures are available in case of needs by the editorial team.

English and style are minor spell check required.

The paper style has been revised.
